# Clinical Remission in Severe Asthma: A Comparative Analysis of Patients with and Without Biologics from the Swiss Severe Asthma Registry

**DOI:** 10.3390/biomedicines13123074

**Published:** 2025-12-12

**Authors:** Fabienne Jaun, Maria Boesing, Giorgia Lüthi-Corridori, Pierre-Olivier Bridevaux, Florian Charbonnier, Christian F. Clarenbach, Jean-Marc Fellrath, Pietro Gianella, Anja Jochmann, Lukas Kern, Nikolay Pavlov, Tsogyal Daniela Latshang, Christophe Von Garnier, Joerg Daniel Leuppi

**Affiliations:** 1University Institute of Internal Medicine, Cantonal Hospital Baselland, 4410 Liestal, Switzerland; 2Medical Faculty, University of Basel, 4056 Basel, Switzerland; 3Center for Executive and Continuing Education, Harvard T.H. Chan School of Public Health—ECPE, Boston, MA 02115, USA; 4Centre Hospitalier du Valais Romand, 1951 Sion, Switzerland; 5University Clinic of Pneumology, University Hospital Geneva, 1205 Geneva, Switzerland; 6Department of Pulmonology, University Hospital Zurich, 8091 Zurich, Switzerland; 7Service of Pulmonology, Réseau Hospitalier Neuchâtelois, 2000 Neuchâtel, Switzerland; 8Pneumology Department, Ente Ospedaliero Cantonale, 6500 Lugano, Switzerland; 9Department of Pneumology, University Children Hospital Basel, 4031 Basel, Switzerland; 10Pneumology Department, Cantonal Hospital Winterthur, 8400 Winterthur, Switzerland; 11Department of Pulmonary Medicine, Allergology and Clinical Immunology Inselspital, Bern University Hospital, University of Bern, 3010 Bern, Switzerland; 12Pneumology Department, Cantonal Hospital Grisons, 7000 Chur, Switzerland; 13Division of Pulmonology, Department of Medicine, Lausanne University Hospital (CHUV) and University of Lausanne, 1011 Lausanne, Switzerland

**Keywords:** severe asthma, clinical remission, asthma remission, real-world evidence

## Abstract

**Background**: Severe asthma is a complex chronic airway disease. Biologic therapies are targeted monoclonal antibody treatments used in patients with uncontrolled, severe asthma, but real-world data from long-term registries and on patients who remain biologic-naïve are limited. This study compared severe asthma patients with and without biologic therapy and identified predictors of key clinical remission components. **Methods**: In this cross-sectional analysis of adult patients from the Swiss Severe Asthma Registry (SSAR), we compared patients treated with a biologic for ≥6 months to biologic-naïve patients (never exposed to biologics). Baseline characteristics were summarized descriptively. Multivariable logistic regression was used to identify predictors of four remission components: good asthma control (ACT ≥ 20), absence of exacerbations, no maintenance oral corticosteroid (OCS) use, and preserved lung function (FEV_1_ > 80% predicted). **Results**: Of 394 patients, 298 (75.6%) were biologic-treated and 96 (24.4%) were biologic-naïve. Biologic-treated patients more often had allergic asthma and type-2–related comorbidities, and showed better outcomes, including fewer exacerbations (0.49 vs. 1.09/year; *p* < 0.001) and higher ACT scores (20.0 vs. 17.2; *p* < 0.001). Biologic therapy was independently associated with higher odds of asthma control (OR 3.96; *p* = 0.006), no exacerbations (OR 5.11; *p* = 0.001), no OCS use (OR 6.27; *p* = 0.002), and FEV_1_ > 80% predicted (OR 4.42; *p* = 0.011). Overall, 24.2% of biologic-treated patients and 6.2% of biologic-naïve patients fulfilled all four remission components. **Conclusions**: In this real-world registry cohort, biologic-treated patients were more likely to meet individual and composite remission criteria than biologic-naïve patients. The relatively low proportion of patients achieving all four stringent criteria highlights the need to revisit current remission definitions and to adopt individualized, multidimensional treatment goals in severe asthma.

## 1. Introduction

Asthma is one of the most common chronic non-communicable diseases worldwide, with over 250 million people affected, and a prevalence rising due to urbanization and lifestyle changes [1,2]. It is associated with both a socio-economic burden as well as a reduced quality of life [3]. Asthma is an inflammatory airway disease and is characterized by increased sensitivity to allergens, pollutants, and airway obstruction [4,5,6]. The severity and frequency of airway symptoms, along with airflow limitation, can vary over time and are influenced by factors such as physical activity, allergens, weather changes, respiratory infections, and comorbid conditions [1,4].

While the majority of asthma cases are mild to moderate, between 3.5 and 10% of patients experience a difficult-to-treat or severe form of the disease. Even though only a minority of patients are suffering from severe asthma, it accounts for a disproportionate amount of the economic burden of asthma, especially asthma-related hospitalization, healthcare costs, with high expenditures on long-term corticosteroid use and biologic therapies [4,7,8,9,10,11,12]. Difficult-to-treat asthma is defined as asthma that remains uncontrolled despite the use of high-dose inhaled corticosteroids (ICS) and oral corticosteroids (OCS), often due to factors such as poor treatment adherence, incorrect inhaler technique, or underlying comorbidities [13]. Severe asthma is a more advanced subset of difficult-to-treat asthma and persists despite optimal treatment and management [13]. It is associated with a high disease burden characterized by frequent recurrent exacerbations, with increased hospitalization rates and persistent airflow limitation [10,13]. Patients with severe uncontrolled asthma often experience a significantly reduced quality of life due to physical activity restrictions, disrupted sleep, or increased psychological distress [7,9,14,15,16].

Severe asthma can further be classified based on the presence or absence of type 2 (T2) inflammation [17,18]. T2 inflammation arises from both innate and adaptive immune responses: environmental triggers such as pollutants or viral and fungal infections activate type 2 innate lymphoid cells (ILC2), whereas allergen exposure stimulates type 2 T-helper (Th2) cells [18,19]. These cells release the key type-2 cytokines—interleukin (IL)-4, IL-5, and IL-13—which drive eosinophilic airway inflammation through enhanced eosinophil maturation, recruitment, and activation [19]. In clinical routine care, T2 biomarkers are typically measured using blood eosinophil counts, fractional exhaled nitric oxide (FeNO), or sputum eosinophils. FeNO reflects IL-13–mediated epithelial inflammation and complements blood eosinophils because it captures airway inflammatory activity that is not always mirrored peripherally [20,21].

Approximately 50–60% of severe asthma cases exhibit a T2-high, eosinophilic phenotype, consistent with observations in Switzerland, whereas the remaining patients present with non-eosinophilic or mixed inflammation [22,23,24]. T2-low asthma is typically characterized by neutrophilic or paucigranulocytic inflammation, is often less responsive to inhaled corticosteroids, and is commonly associated with late-onset disease [22]. Mechanistically, T2-low asthma is linked to activation of Type 1 (Th1) or Type 17 (Th17) pathways rather than T2 cytokine signaling, resulting in airway inflammation not driven by eosinophils [19,22,24]. Distinguishing between T2-high and T2-low phenotypes is clinically essential, particularly for selecting biologic therapies that target specific inflammatory pathways [24,25,26].

The eosinophilic phenotype accounts for approximately 50 to 60%, whereas the rest is classified as either non-eosinophilic or a mixed form of severe asthma cases, which aligns with previous observations in Switzerland [23,24,27]. The distinction between the T2-high and T2-low phenotypes is clinically important, especially for selecting biologic therapies targeting those inflammatory pathways [26].

### 1.1. Current Treatment Landscape

Asthma treatment follows a stepwise approach, with patients with severe asthma receiving medium- to high-dose ICS combined with long-acting beta-agonists (LABA), with or without long-acting muscarinic antagonists (LAMA) [13]. However, despite these standard therapies, a substantial proportion of patients with severe asthma fail to achieve adequate disease control, remain dependent on OCS and/or experience recurrent exacerbations [25,28,29,30].

Standard asthma treatments present challenges, particularly due to the long-term adverse effects of OCS, including osteoporosis, increased cardiovascular risk, and adrenal suppression [30,31,32]. For patients who are not reaching disease control, more targeted approaches are necessary.

#### The Role of Biologic Agents in Severe Asthma

Biologic therapies are monoclonal antibody treatments designed to target specific components of the immune pathways that drive severe asthma. Unlike conventional therapies, which have broad anti-inflammatory effects, biologics selectively inhibit key mediators such as interleukin (IL)-5, IL-4, IL-13, immunoglobulin E (IgE), or thymic stromal lymphopoietin (TSLP) [33,34]. These mediators play central roles in type 2 inflammation: IL-5 promotes eosinophil maturation and survival; IL-4 and IL-13 drive IgE class switching, mucus production, and airway hyperresponsiveness; IgE mediates allergic sensitization and mast-cell activation; and TSLP acts as an upstream epithelial cytokine that amplifies multiple inflammatory pathways [33].

Several biologics are approved in Europe and Switzerland, including omalizumab (anti-IgE), mepolizumab (anti-IL-5), reslizumab (anti-IL-5), benralizumab (anti-IL-5Rα), dupilumab (anti-IL-4Rα, inhibiting IL-4/IL-13 signaling), and tezepelumab (anti-TSLP). These targeted agents have demonstrated significant clinical benefits, including reductions in exacerbation rates, improvements in lung function, and decreased dependence on systemic corticosteroids [10,33,35].

New emerging therapies, including mesenchymal stromal cell–based approaches and gene-targeted strategies, are being explored for future treatment; however, these remain experimental. Biologic therapy, therefore, represents the only widely available, evidence-based targeted treatment option for severe asthma at present [36,37].

In Switzerland, biologic therapies are reimbursed under strict criteria, including documented severe asthma with evidence of T2 inflammation—such as elevated blood eosinophils, FeNO, or IgE—frequent exacerbations, and persistent symptoms despite optimized high-dose inhaled therapy [33,38]. Similar eligibility criteria exist across Europe [39].

### 1.2. Study Rationale

Despite the growing use of biologics in severe asthma management, several knowledge gaps remain. Most studies primarily assess short-term outcomes, while real-world data on the long-term effectiveness of biologics (≥12 months) are limited [33]. The durability of treatment benefits, including sustained exacerbation reduction, lung function improvement, and corticosteroid-sparing effects, requires further investigation.

Despite advancements in asthma treatment, there is limited understanding of why some patients achieve disease control without biologic therapy. Factors such as natural disease progression, unidentified biomarkers, variations in immune response, or differences in treatment adherence and lifestyle may play a role [27,40,41,42]. However, real-world data on these patients remain scarce, highlighting the need for further research to identify predictors of asthma control independent of biologics.

Finally, one of the primary goals of biologic therapy is to reduce dependence on OCS, which are associated with significant long-term side effects [31,32,43,44]. Although clinical trials have shown a reduction in OCS use with biologics, real-world evidence on the extent of steroid-sparing benefits and their sustainability over time remains limited [45,46,47,48]. Addressing these gaps through robust cohort analyses will provide critical insights into optimizing severe asthma treatment strategies.

### 1.3. Study Objectives

The aims of this study are to compare patients with severe asthma who have received biologic treatment for at least six months to those who are biologic-naïve, and to identify predictors of asthma remission (defined as asthma control test (ACT) score ≥ 20), no OCS use, absence of exacerbations, and preserved lung function (defined as forced expiratory volume in one second (FEV_1_) > 80% predicted). However, given the cross-sectional design, this study can identify associations but cannot establish causal relationships between biologic therapy and clinical outcomes.

## 2. Materials and Methods

### 2.1. Study Design and Population

This study is a cross-sectional analysis nested within the Swiss Severe Asthma Registry (SSAR), which is an ongoing, multicenter, prospective, open cohort study initiated in April 2019. Patients are enrolled from primary, secondary, and tertiary care facilities across Switzerland. Eligibility criteria for inclusion are: diagnosis of severe asthma as defined by the European Respiratory Society (ERS)/American Thoracic Society (ATS) guidelines, age ≥ 6 years, and treatment in accordance with GINA steps 4–5 [13,49]. Patients with a life expectancy <6 months or inadequate language proficiency (German, French, or Italian) are excluded.

For this study, we included adult patients (≥18 years) from the registry who had either been on biologic therapy for at least six months or were biologic-naïve (never treated with a biologic) and were included between April 2019 and March 2025 (Figure 1).

Data were collected and entered in a non-public electronic register provided by the German Asthma Net e.V., located in Mainz, Germany [50].

### 2.2. Outcomes

The outcomes used in this study represent a four-component definition of clinical remission, as proposed by Lommatzsch and Virchow (2025) [50]. The criteria are asthma control, defined as an ACT score ≥ 20, no use of OCS as a maintenance medication, preserved lung function, defined as FEV_1_ > 80% predicted, and the absence of exacerbations, defined as requiring OCS for ≥3 days and/or leading to emergency department visits or hospitalizations for asthma in the previous year. ACT scores range from 5 to 25, with scores ≥20 indicating well-controlled, 16–19 partly controlled, and ≤15 poorly controlled asthma.

Clinically, an ACT score ≥ 20 reflects low symptom burden (fewer day-to-day symptoms, improved sleep, and better activity tolerance) and a reduced risk of future exacerbations. The absence of maintenance OCS use reflects adequate disease control without dependence on systemic steroids. Achieving an FEV_1_ > 80% predicted indicates preserved airflow and is associated with improved functional status and long-term prognosis. The absence of exacerbations indicates stable disease without acute worsening, requiring intensified treatment.

Because remission is defined as a sustained state over ≥12 months, our cross-sectional analysis captures only the presence of remission components at a single time point. Therefore, findings should be interpreted as the likelihood of meeting individual remission criteria rather than as confirmed longitudinal remission.

### 2.3. Statistical Analysis

All analyses were conducted using R version 4.3.1 (16 June 2023). Implausible values were identified and excluded if critical.

#### 2.3.1. Missing Data

Missing data were assumed to be missing at random (MAR). A tiered imputation strategy was implemented. For variables with minimal missing data (<5% for categorical, <10% for continuous), missing values were imputed using mode and median imputation, respectively. Variables with more complex patterns of missing data were left as missing. Sensitivity to imputation strategies was assessed via a complete case analysis. The variables that were imputed and their extent of missing data can be found in Appendix A.

#### 2.3.2. Descriptive Statistics

Continuous variables were summarized using means and standard deviations (SD); in case of non-normal distribution, the geometric mean and geometric standard deviation (GSD) were used. Categorical variables were summarized as counts and proportions. These summaries were stratified by biologic treatment status.

Comparisons between biologic-treated (BT) and biologic-naïve (nBT) patients were performed using the Wilcoxon rank-sum tests for continuous variables and Chi-square tests or Fisher’s exact tests for categorical variables.

#### 2.3.3. Regression Modeling

Multivariable logistic regression models with robust standard errors were used to find predictors for the four components of asthma remission as described in Section 2.2. The results are reported with Odds Ratios (OR) and 95% confidence intervals (CIs). All models included the following predictors: age, sex, body mass index (BMI), time of asthma onset, comorbidities (chronic obstructive pulmonary disease (COPD), gastro esophageal reflux disease (GERD), nasal polyps, depression), biomarkers (BEC, FeNO, lowest FEV_1_% predicted (last 2 years)), use of high-dose ICS, and a positive family history of asthma. The predictors were chosen according to the recent literature [27,51,52,53].

## 3. Results

A total of 524 patients were included in the registry between April 2019 and April 2025 and assessed for eligibility. After excluding patients who did not fulfill the inclusion criteria (≥18 years of age, currently on biologic therapy for at least six months, or biologic-naïve (never treated with a biologic) 394 patients were included in the analysis, of whom 96 (24.4%) were biologic-naïve (nBT) and 298 (75.6%) had received biologic treatment for at least 6 months(BT) (Figure 1).

### 3.1. Descriptive Analysis

The proportion of females was slightly higher (BT: 44.0% vs. nBT: 53.%, *p* = 0.15) and mean age slightly lower (BT: 55.6 vs. nBT: 52.9 years, *p* = 0.13) in biologic-treated patients, when compared to biologic-naïve ones, though these differences were not statistically significant (Table 1). BMI and weight category distribution were similar in both groups, with no significant difference (Table 1).

Among BT patients, the mean duration of biologic therapy was 27.8 months with a range between 6 and 140 months, and the highest BEC prior to treatment was 420 cells/μL (median). Most patients received Mepolizumab (33.2%) followed by Benralizumab (27.5%), Omalizumab (22.5%), and Dupilumab (15.8%). For 67.8% of the BT patients, it was their first biologic (Figure 1).

Allergic asthma was significantly more prevalent in the BT group (BT: 49.3% vs. nBT: 34.4%, *p* = 0.02), while non-allergic asthma was more common in nBT patients (BT: 32.2% vs. nBT: 44.8%). Childhood-onset asthma and family history of asthma were similarly distributed across both groups.

Influenza vaccination uptake was significantly higher among BT patients (BT: 60.5% vs. nBT: 46.3%, *p* < 0.01), while pneumococcal vaccination rates were comparable (BT: 24.7% vs. nBT: 24.6%, *p* = 0.84). Smoking status also differed significantly (*p* = 0.02), with current smoking being more common in the nBT (nBT: 15.6% vs. BT: 7.0%).

Indicators of disease burden showed clear differences: nBT patients had more exacerbations on average (nBT: 1.70 vs. BT: 0.86, *p* < 0.001), lower ACT scores (BT: 19.98 vs. nBT: 17.23, *p* < 0.001), and a markedly lower proportion of patients without respiratory symptoms (nBT: 4.2% vs. BT: 23.2%, *p* < 0.001).

Use of high-dose ICS was more frequent in BT patients (BT: 89.4% vs. nBT: 78.2%, *p* = 0.01), though the mean ICS dose was similar (BT: 1345 vs. nBT: 1417 μg/day, *p* = 0.33). OCS use did not differ in prevalence (17–18%), but nBT patients were taking a significantly higher mean daily prednisolone equivalent dose (nBT: 14.4 vs. BT: 11.6 mg/day, *p* = 0.03).

Pulmonary function was generally better in BT patients with higher FEV_1_% predicted (BT: 77.6% vs. nBT: 72.3%, *p* < 0.01) and forced vital capacity (FVC) % predicted (BT: 90.9% vs. nBT: 86.3%, *p* = 0.01). The lowest FEV_1_ in the last two years and bronchial hyperreactivity (BHR) rates were similar.

Regarding remission criteria, BT patients were significantly more likely to fulfill an individual criterion of no exacerbations (BT: 59.7% vs. nBT: 36.5%, *p* < 0.001), ACT score ≥ 20 (BT: 65.5% vs. nBT: 33.7%, *p* < 0.001), and FEV_1_ > 80% predicted (BT: 44.3% vs. nBT: 27.1%, *p* < 0.01). Only for the non-use of OCS, the proportions were similarly high in both groups (BT: 82.6% vs. nBT: 82.3%, *p* = 1). The Venn diagram (Figure 2, Appendix A) showed that 24.2% of BT patients fulfilled all four criteria compared with 6.2% of biologic-naïve patients. Using the less stringent remission definition (ACT-controlled, no exacerbations, no OCS), 45.7% of BT patients met the criteria versus 15.6% in the naïve group (Figure 2, Appendix A).

Co-morbid conditions such as nasal polyps (BT: 39.4% vs. nBT: 15.1%, *p* < 0.001) and sinusitis (BT: 48.1% vs. nBT: 35.5%, *p* = 0.04) were significantly more common in BT patients. Other comorbidities, including GERD, depression, COPD, eosinophilic granulomatosis with polyangiitis (EGPA), allergic bronchopulmonary aspergillosis (ABPA), and bronchiectasis, were comparable between groups (Table 1, Appendix A).

### 3.2. Multivariable Analysis

Multivariable logistic regression was used to identify factors associated with the remission criteria: asthma control (ACT score ≥ 20), OCS use, absence of exacerbations, and a FEV_1_ > 80% predicted.

#### 3.2.1. Asthma Control

Biologic treatment status emerged as a significant predictor for asthma control: BT patients had significantly higher odds for having well-controlled asthma compared to nBT patients (OR = 3.96; 95% CI: 1.48–10.56; *p* = 0.006) (Figure 3). A history of switching biologic therapies (“Biologic Switcher”) was associated with lower odds of asthma control (OR = 0.39; 95% CI: 0.17–0.88; *p* = 0.023).

The presence of comorbid COPD showed a non-significant trend towards lower odds of having controlled asthma (OR = 0.12; 95% CI: 0.020–1.014; *p* = 0.055). Other variables did not show statistically significant associations with asthma control (*p* > 0.05 for all).

Notably, while allergic comorbidity (OR = 0.46; *p* = 0.070) showed a trend, it did not reach statistical significance. These findings highlight that biologic therapy is independently associated with better asthma control, even when accounting for a broad set of demographics, clinical, and pulmonary variables.

#### 3.2.2. No OCS Use

BT patients have significantly better odds of OCS non-use compared to nBT patients (OR = 6.27; 95% CI: 1.98–19.81; *p* = 0.002) (Figure 4). Patients with a history of biologic switching showed significantly lower odds of achieving OCS independence (OR = 0.20; 95% CI: 0.06–0.67; *p* = 0.009).

Only one other variable emerged as a statistically significant predictor of OCS non-use. A higher BMI was associated with increased odds of not using OCS (OR = 1.14; 95% CI: 1.04–1.26; *p* = 0.009). In addition, a higher lowest FEV_1_% predicted in the past 2 years showed a non-significant association with OCS non-use (OR = 1.02; 95% CI: 1.0–1.051; *p* = 0.055). The other variables in the model were not significantly associated with OCS non-use in this model (*p* > 0.05 for all).

#### 3.2.3. FEV1 > 80% Predicted

BT patients had significantly higher odds of achieving normal lung function compared to those treated with biologics (OR = 4.42; 95% CI: 1.39–14.06; *p* = 0.011) (Figure 5). GERD was significantly associated with reduced odds of normal lung function (OR = 0.18; 95% CI: 0.05–0.67; *p* = 0.010). COPD showed a borderline negative association (OR = 0.22; *p* = 0.061), suggesting potential clinical relevance but not reaching statistical significance. The most robust predictor of FEV_1_ > 80% predicted was the lowest FEV_1_% predicted in the past 2 years, with higher values significantly increasing the odds of current normal lung function (OR = 1.14; 95% CI: 1.09–1.19; *p* < 0.001). The other variables in our model were not significantly associated with a current FEV_1_% predicted > 80%.

#### 3.2.4. Absence of Asthma Exacerbations

Being treated with a biologic was significantly associated with increased odds of experiencing no exacerbations compared to patients receiving no biologic therapy (OR = 5.11; 95% CI: 2.01–12.99; *p* = 0.001) (Figure 6). Depression showed a borderline positive association with being free of exacerbations (OR = 3.40; 95% CI: 0.96–12.04; *p* = 0.057), suggesting a potential—but statistically inconclusive—protective effect. Conversely, COPD was associated with lower odds of being exacerbation-free, although this did not reach statistical significance (OR = 0.43; *p* = 0.216). The other variables were not significantly associated with the absence of exacerbations (*p* > 0.05 for all).

### 3.3. Sensitivity Analysis

To evaluate the robustness of our findings and the impact of missing data, we conducted a sensitivity analysis comparing model performance using imputed versus non-imputed (complete case) datasets across four key binary outcomes: asthma control (ACT score ≥ 20), no exacerbations, no OCS use, and FEV_1_ > 80% predicted. The sensitivity analyses on the complete-case dataset demonstrated that data imputation did not meaningfully alter model estimates. Across all four outcomes, ORs from complete-case models remained within the 95% confidence intervals of the imputed models, and no changes in statistical significance were observed (Appendix A).

## 4. Discussion

In the current study, we analyzed data from the Swiss Severe Asthma Registry, exploring the difference between BT patients who were treated with a biologic agent for at least 6 months and nBT patients, who were never treated with a biologic. In addition, we explored predictors of four outcomes as components of clinical asthma remission [50]: asthma control (defined as ACT score ≥ 20 points), absence of exacerbations, no OCS use, and preserved lung function (FEV_1_ > 80% predicted). Several key findings emerged, offering insights into disease control and influencing factors.

### 4.1. Differences Between Biologic-Naïve and Biologic-Treated Patients

The comparative analysis highlights several key differences between BT and nBT severe asthma patients.

More than 50% of patients in both groups are either overweight or obese. This aligns with previous findings about the prevalence of overweight and obesity in similar severe asthma cohorts [54,55,56]. Obesity is known to be associated with a higher burden of disease as well as with worse disease control [54,55,57]. The proportion of patients who are overweight (BMI ≥ 25 but <30) was similar in both groups (38.5% vs. 35.8%); however, the proportion of patients who are obese is higher in the nBT group (31.2% vs. 24.3%). This difference might contribute to worse disease control and a higher burden of disease in the biologic-naïve group.

Allergic asthma was more prevalent in the group treated with biologics compared to the biologic-naïve group (49.3% vs. 34.4%). This could be partially explained by the fact that currently available biologics target specific mechanisms, such as IgE or interleukin pathways, which are associated with allergic or eosinophilic asthma phenotypes [33,58].

We observed a significantly higher vaccination rate for influenza in the biologic-treated group (60.5% vs. 46.3%), despite being recommended for all patients with severe asthma [59,60,61]. Overall, this gap in vaccination coverage is concerning. The higher rates of influenza vaccination in biologic-treated patients with severe asthma might reflect better awareness and/or engagement for preventive measures on both the patients’ and providers’ side in this group [62,63,64]. However, a difference in the vaccination rates for pneumococcus was not observed.

A higher proportion of biologic-naïve patients were current smokers compared to the biologic-treated group (15.6% vs. 7%, *p* = 0.02). This could possibly be explained by lower treatment adherence in current smokers, which subsequently could lead to fewer biologic prescriptions, despite the fact that smokers would equally benefit from biologic treatment [65,66]. These findings are consistent with an analysis from the International Severe Asthma Registry (ISAR), showing that patients beginning biologic treatment are less likely to be smokers compared to biologics-naïve patients [67].

The disease burden was significantly lower in the biologic-treated group. Biologic-treated patients had fewer exacerbations, better asthma control, fewer symptoms, and better lung function. Although causality cannot be established due to the cross-sectional design, the consistent and substantial differences across multiple independent clinical outcomes are consistent with a beneficial real-world effect of biologic therapy in a difficult-to-treat population [68,69,70]. Notably, the proportion of patients on OCS did not differ between groups, yet biologic-treated patients required significantly lower daily doses. Biologics are well known for their steroid-sparing effect, which is an important treatment goal, given the side effect profile of long-term OCS use [46,47,70,71]. The incomplete OCS withdrawal in patients treated with biologics may be influenced by comorbidities and by physicians’ caution, as the duration of biologic treatment in the observation period was too short [72,73].

Burden of disease may not be the only driver of initiating biologic treatment, as a higher proportion of biologic-naïve patients had poor disease control in our cohort. This might also reflect inequities in access to biologic therapy, inadequate eligibility criteria, hesitancy for treatment initiation, or adherence issues. Overall, our findings raise the critical question of whether biologics are underutilized in symptomatic patients with severe asthma.

There was a significantly higher prevalence of nasal polyps and sinusitis in the biologic-treated group, which indicates an overlap of T2 inflammation-driven diseases that might reinforce the decision to initiate a biologic treatment.

In summary, biologic-treated patients showed better asthma control and lung function, highlighting the real-world effectiveness of those treatments [38,68]. Differences in comorbidities, smoking, and vaccination status suggest potential barriers to biologic access. These findings support the need for a tailored, multidisciplinary approach in managing severe asthma [74,75].

### 4.2. Predictors for Asthma Remission

The concept of remission in severe asthma is emerging due to the established biologic treatments and a new personalized approach. Asthma remission moves the treatment goals beyond symptom management and toward achieving sustained disease control and reduction in corticosteroid dependence [10,52,53]. Biologic treatment can improve severe asthma substantially and lead to a (partial) remission in a subset of patients [76,77,78,79].

In this study, we evaluated predictors of four remission criteria in severe asthma, as proposed by Lommatzsch and Virchow (2025)—namely, asthma control, absence of exacerbations, no OCS use, and FEV_1_ > 80% predicted [50]. In line with currently published remission frameworks, all four components were equally weighted, as no validated approach to differential weighting of remission criteria in severe asthma is available. However, Porsbjerg and colleagues have emphasized that achieving remission requires more than symptom suppression; they suggested a more patient-tailored and domain-based approach, one that targets the right disease mechanisms at the right time while acknowledging that individual patients may differ in their ability to meet each criterion at any time point [80]. Patients receiving biologic therapy demonstrated significantly higher odds of achieving the individual remission criteria compared with biologic-naïve patients. In our cohort, one in five biologic-treated patients fulfilled all four remission criteria, whereas only one in seventeen biologic-naïve patients did. Although this demonstrates that full remission is achievable in a subset of patients, the majority did not reach all components simultaneously, highlighting the stringency of the current remission framework [10,52,77,81].

Therefore, achieving all four remission components simultaneously remains clinically challenging. A recent analysis from the UK Severe Asthma Registry reported clinical remission in 25% of patients at 9–24 months and 32% at 30–48 months, allowing for biologic switching and using a three-component definition that did not include lung function [82]. Likewise, long-term data from Denmark evaluating anti-IL5/Rα biologic therapy showed that although 37% of patients achieved remission during at least one year of follow-up, only 7.7% maintained sustained five-year remission when all components, including lung function, were considered [83]. Together with recent conceptual work emphasizing that remission is an ambitious but attainable treatment goal, these findings highlight that remission is a dynamic and often unstable state, and that meeting all criteria concurrently remains difficult in real-world severe asthma [84].

The four-component remission definition may be overly stringent for certain patient groups. For example, achieving FEV_1_ > 80% predicted may be unrealistic in older patients with long-standing airway remodeling. ‘No OCS use’ does not distinguish between successful tapering and patients who were never steroid-dependent. These aspects underscore the need to refine remission definitions for real-world applicability [85,86].

These findings must be interpreted in the context of our cohort characteristics, with a mean treatment duration of 27 months and a mean age of 56 years, of whom around 60% were overweight or obese. Older age, longer asthma duration, and elevated BMI are well-established barriers to remission, as they are associated with more fixed airflow obstruction, increased comorbidity burden, and diminished responsiveness to biologic therapy [10,52,77,81,87].

In addition, our cohort also included patients who had already switched biologics one or even more times. Switching biologics can improve asthma control, reduce exacerbations, and lower oral corticosteroid use [88,89]. Nonetheless, achieving all four remission criteria—especially FEV_1_ > 80% predicted—is less likely in switchers than in initial responders, largely due to persistent airflow limitation, longer disease duration, and comorbidities such as obesity and depression [10,88,89]. Our analysis showed that patients who switched biologics are significantly less likely to achieve asthma control (Figure 3) or be free of OCS (Figure 4). Evidence on remission after biologic switching remains limited, with few prospective studies and no consensus on standardized remission definitions. Variability in criteria, particularly strict lung function thresholds, further complicates comparisons and highlights the need for harmonized approaches [10,88,90]. Therefore, we assume that patients who switch biologics often represent a more severe or treatment-refractory subgroup, which may partially explain the observed differences.

To improve outcomes, emphasis should be placed on early intervention, including a comprehensive diagnostic, targeting modifiable traits, and adopting comprehensive management strategies [52,91]. As recommended by the Global Initiative for Asthma, remission should be viewed as a realistic yet ambitious treatment goal. Future research should refine remission criteria, optimize timing of biologic initiation, and address risk factors such as obesity and comorbidities [92].

Although biologic therapies provide substantial clinical benefit towards the improvement of severe asthma, their high cost and restricted reimbursement criteria limit access in many low- and middle-income countries [39,93,94]. As a result, the generalizability of our findings to settings without universal access to biologics is limited, and the global policy impact must be interpreted with caution.

#### 4.2.1. Asthma Control

Achieving an ACT score ≥ 20 represents a clinically meaningful threshold, as it corresponds to well-controlled asthma with fewer symptoms and improved quality of life [27,95].

Biologic-naïve patients as well as patients who switched biologics have significantly lower odds of achieving asthma control, defined as an ACT score ≥ 20 points. These finding highlights the challenges in asthma management that this subgroup of patients face.

While other variables, such as COPD, allergies, depression, and family history, showed trends toward lower odds, they did not reach statistical significance. Those findings are similar to a previous study of this cohort and to findings from other cohort studies [27,51,79,95,96]. Interestingly, biological markers like BEC and prior FEV_1_% predicted did not significantly predict asthma control, suggesting that other clinical features may play a more dominant role in this context than physiological features [52,53].

#### 4.2.2. No OCS Use

The proportion of patients requiring maintenance OCS did not differ between groups, yet in the multivariable analysis, biologic treatment was associated with higher odds of being OCS-free, whereas biologic switching was associated with lower odds of being OCS-free, which is in line with previous findings on the steroid-sparing effect of biologics [46,47,72]. However, this discrepancy suggests the presence of confounding factors. For example, coexisting conditions such as COPD. COPD may influence the likelihood of OCS discontinuation, as patients with COPD might be less able to taper off corticosteroids [97,98]. It has been shown that obesity related asthma is less responsive to corticosteroids, which may explain why patients with a higher BMI have higher odds of being OCS-free [54,56,57]. Alternatively, the awareness of weight gain as a common side effect of maintenance OCS use and the association of obesity with a higher burden of disease leads to better OCS withdrawal/reduction strategies in this particular population [99]. In our analysis, past lung function (lowest FEV_1_% predicted in the past two years) showed a trend of increased odds of achieving the no OCS use criterion of asthma remission. Preserved pulmonary function may indicate an earlier diagnosis and, therefore, faster treatment escalation to improve the overall outcome [68]. Our observations highlight that achieving the criteria for being OCS-free is influenced by multiple patient-level factors beyond those included in our model.

#### 4.2.3. FEV_1_ > 80% Predicted

From a clinical standpoint, attaining an FEV_1_ above 80% predicted indicates preserved lung function, which translates into reduced breathlessness, better exercise capacity, and a lower likelihood of progressive fixed airflow obstruction [100,101].

Biologic-naïve status was associated with significantly lower odds of achieving a FEV_1_ > 80% predicted. The strongest predictor for a FEV_1_ > 80% predicted is the lowest FEV_1_% predicted in the last two years. This finding reinforces lung function as a key determinant for future lung function stability, as shown before [68,100]. Other comorbidities and demographic factors did not show a significant association with FEV_1_ > 80% predicted, suggesting that physiological markers may be more informative than clinical history alone in predicting preserved lung function. In contrast to other studies, BEC was not associated with FEV_1_ > 80% predicted in our cohort, probably due to the small sample size [102,103,104,105].

#### 4.2.4. Absence of Asthma Exacerbations

Although biologic-naïve status was strongly associated with a higher risk of exacerbations in the present analysis, comorbidities and demographic factors did not emerge as independent predictors. This contrasts with prior analyses from the same cohort, where gastroesophageal reflux disease and multimorbidity were significantly linked to exacerbation risk [51,106]. The difference between the current and previous analyses may be explained by the differences in study design (longitudinal vs. cross-sectional) and the sample size, as well as the inclusion of different covariates. Nonetheless, extensive evidence demonstrates that comorbidities such as chronic rhinosinusitis, nasal polyposis, and gastroesophageal reflux disease increase exacerbation risk in severe asthma [51,106,107,108,109]. Reducing exacerbations remains an important treatment goal, as recurrent exacerbations affect quality of life, increase the burden of disease, and have a significant impact on direct, indirect, and intangible asthma-related costs [110,111,112]. Those findings reinforce the real-world impact of biologics while highlighting the complex interplay between severe asthma, patient characteristics, and other influencing factors [113,114].

### 4.3. Limitations

Data on socioeconomic status, medication adherence, and incorrect inhalation technique, which might influence asthma-related outcomes, have not been collected. Due to the study design, reporting errors cannot be excluded. In addition, we must assume a selection bias, as biologic-treated patients or those with good treatment and appointment adherence have been preferentially recruited [115,116]. The dichotomous reporting of several variables also limited the depth of analysis. Finally, because exposures and outcomes were assessed at a single time point, the cross-sectional design precludes establishing temporality and therefore does not permit causal interpretation. In addition, biologic switching reflects refractory disease and may confound comparisons between groups; although switching status was recorded, the study was not powered for stratified analyses, and residual confounding cannot be excluded. Because outcomes were assessed at a single time point, temporal relationships cannot be established, and the observed associations may be confounded by factors such as disease duration or prior treatment response.

## 5. Conclusions

This analysis of the Swiss Severe Asthma Registry demonstrates that biologic-treated patients exhibit substantially better clinical outcomes than biologic-naïve individuals, including fewer exacerbations, higher ACT scores, and improved lung function, reinforcing the robust real-world effectiveness of biologic therapies as a central component of personalized severe asthma management.

Importantly, approximately one in five biologic-treated patients achieved all four remission criteria, showing that full remission is attainable in a subset of patients. However, full remission remained uncommon, underscoring that the current four-component definition, particularly the requirement of FEV_1_ > 80% predicted, may be overly stringent for certain patient groups, such as older individuals or those with long-standing airway remodeling. This highlights the need to refine remission definitions or apply them in a more individualized manner.

Our findings further reveal potential inequities in access or eligibility for biologics and underscore the influence of comorbidities, lung function history, and disease severity, including biologic switching, on treatment success. These observations emphasize the importance of early intervention, careful phenotyping, and multidisciplinary management strategies.

Overall, this study provides important real-world evidence supporting biologic therapy as the most effective currently available option for achieving meaningful clinical improvement in severe asthma. At the same time, it underscores the need to improve access, optimize patient selection, and refine remission frameworks so that remission becomes a realistic and individualized treatment goal in clinical practice.

## Figures and Tables

**Figure 1 biomedicines-13-03074-f001:**
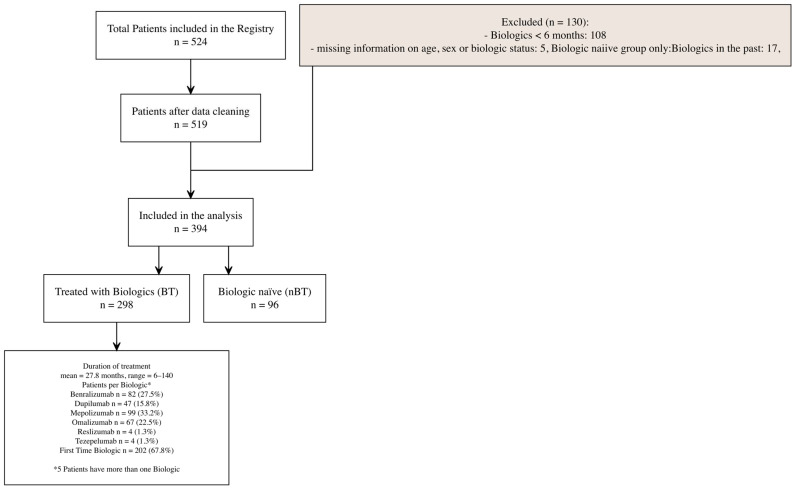
Patient flowchart, showing patient selection, inclusion/exclusion criteria, and categorization into biologic-treated and biologic-naïve groups.* 5 Patients have more than one biologic.

**Figure 2 biomedicines-13-03074-f002:**
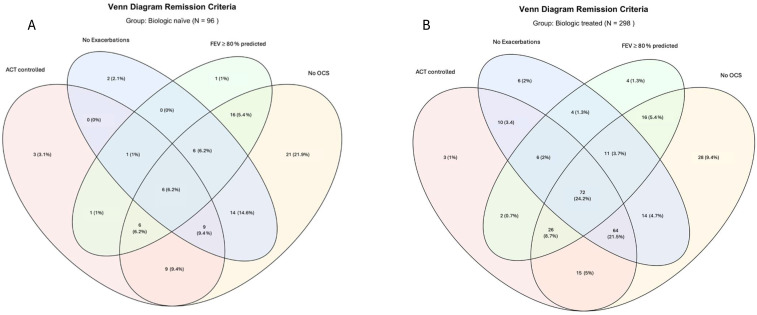
Venn Diagrams of the remission criteria for the biologic naïve group (**A**) and the biologic treated group (**B**).

**Figure 3 biomedicines-13-03074-f003:**
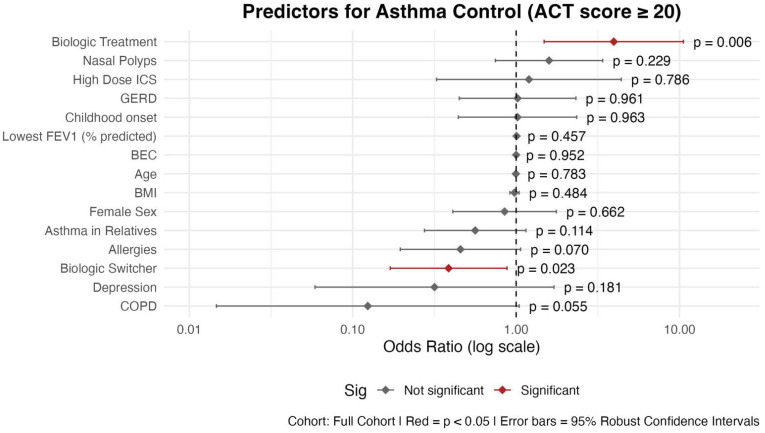
Predictors of asthma control (ACT score ≥ 20): logistic regression results shown as OR with 95% confidence intervals. Abbreviations: BEC: Blood Eosinophil Count; BMI: Body Mass Index; COPD: Chronic Obstructive Pulmonary Disease; GERD: Gastroesophageal Reflux Disease; ICS: Inhaled Corticosteroids; Childhood onset: before the age of 12. Asthma in first-degree relatives (mother, father, siblings, and own children). Model Specifications: GLM; family: binomial; link function: logit; estimation: maximum likelihood with robust standard errors. Outcome: ACT controlled, defined as ACT score ≥ 20.

**Figure 4 biomedicines-13-03074-f004:**
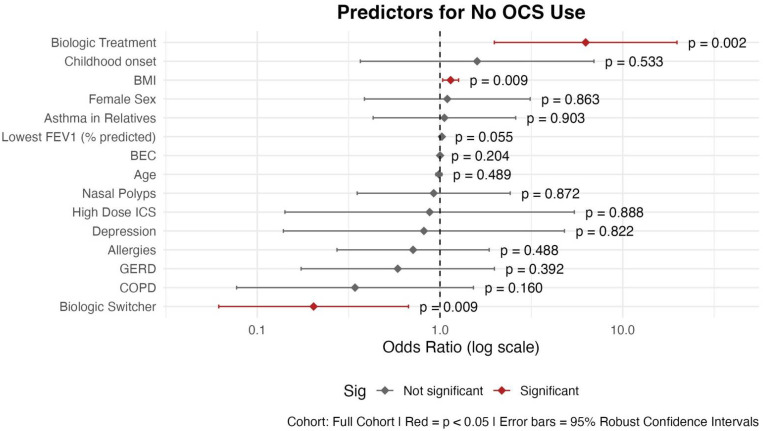
Predictors of no OCS use: logistic regression results shown as OR with 95% Confidence intervals. Abbreviations: BEC: Blood Eosinophil Count; BMI: Body Mass Index; COPD: Chronic Obstructive Pulmonary Disease; GERD: Gastroesophageal Reflux Disease; ICS: Inhaled Corticosteroids; OCS: Oral Corticosteroids. Childhood onset: before the age of 12. Asthma in first-degree relatives (mother, father, siblings, and own children). Model Specifications: GLM; family: binomial; link function: logit; estimation: maximum likelihood with robust standard errors. Outcome: No OCS use.

**Figure 5 biomedicines-13-03074-f005:**
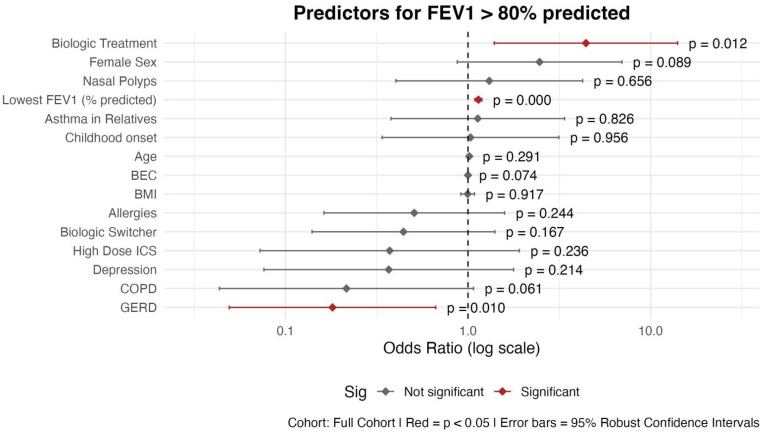
Predictors for FEV_1_ > 80% predicted: logistic regression results shown as OR with 95% confidence intervals. Abbreviations: BEC: Blood Eosinophil Count; BMI: Body Mass Index; COPD: Chronic Obstructive Pulmonary Disease; GERD: Gastroesophageal Reflux Disease; ICS: Inhaled Corticosteroids; Childhood onset: before the age of 12. Asthma in first-degree relatives (mother, father, siblings, and own children). Model Specifications: GLM; family: binomial; link function: logit; estimation: maximum likelihood with robust standard errors. Outcome: FEV_1_ > 80% predicted.

**Figure 6 biomedicines-13-03074-f006:**
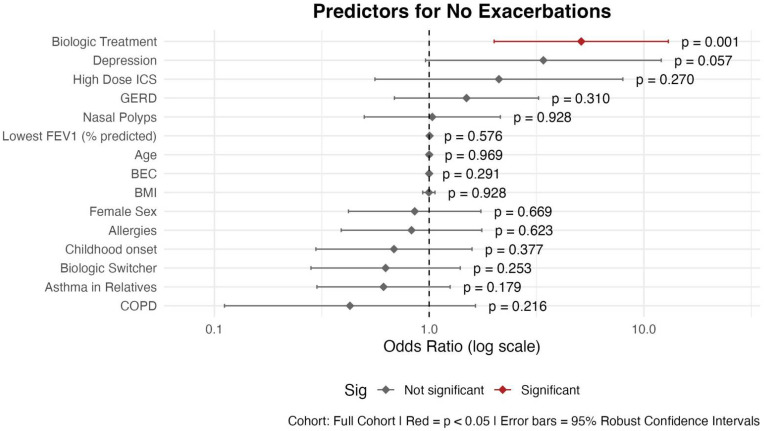
Predictors for the absence of asthma exacerbations: Logistic Regression Results with Robust Standard Errors. Abbreviations: BEC: Blood Eosinophil Count; BMI: Body Mass Index; COPD: Chronic Obstructive Pulmonary Disease; GERD: Gastroesophageal Reflux Disease; ICS: Inhaled Corticosteroids; Childhood onset: before the age of 12. Asthma in first-degree relatives (mother, father, siblings, and own children). Model Specifications: GLM; family: binomial; link function: logit; estimation: maximum likelihood with robust standard errors. Outcome: Absence of exacerbations.

**Table 1 biomedicines-13-03074-t001:** Patient characteristics in both groups.

	Biologic-Naïve (*n* = 96)	Biologic-Treated (*n* = 298)	*p*-Value
Female, *n* (%)	51 (53.1%)	131 (44%)	0.15
Age, Mean (SD)	52.92 (16.02)	55.63 (15.67)	0.13
BMI, Mean (SD)	26.86 (5.73)	27.85 (5.71)	0.06
Underweight, *n* (%)	5 (5.2%)	8 (2.7%)	0.11
Normal weight, *n* (%)	24 (25%)	110 (37.2%)	
Overweight, *n* (%)	37 (38.5%)	106 (35.8%)	
Obese, *n* (%)	30 (31.2%)	72 (24.3%)	
**Type of Asthma**			
Allergic, *n* (%)	33 (34.4)	147 (49.3)	0.02
Non-Allergic, *n* (%)	43 (44.8)	96 (32.2)	
Mixed, *n* (%)	20 (20.8)	55 (18.5)	
**Smoking Status**			
Lifelong Non-Smoker, *n* (%)	52 (54.2)	159 (53.4)	0.02
Former-Smoker, *n* (%)	29 (30.2)	118 (39.6)	
Current Smoker, *n* (%)	15 (15.6)	21 (7)	
**Disease Burden**			
ACT Score, mean (SD)	17.23 (4.45)	19.98 (4.68)	<0.001
No Symptoms, *n* (%)	4 (4.2)	69 (23.2)	<0.001
**Exacerbations in the previous year**			
Number of Exacerbations, geom. mean (GSD) ^b^	1.09 (2.02)	0.49 (1.80)	<0.001
No Exacerbation	38 (39.6)	187 (62.8%)	<0.001
≤2 Exacerbations per year	32 (33.3)	81 (27.2%)	
>2 Exacerbations per year	26 (27.01)	30 (10.1%)	
**Medication**			
High-dose ICS, *n* (%)	68 (78.2%)	253 (89.4%)	0.01
Beclomethasone Equivalent (μg/day), Mean (SD)	1417 (786)	1345 (873)	0.33
OCS, *n* (%)	17 (17.9)	52 (17.4)	1
Prednisolone Equivalent (mg/day), Mean (SD)	14.4 (12.8)	11.6 (15)	0.03
**Pulmonary Function**			
FEV_1_ (% predicted), Mean (SD)	72.33 (19.55)	77.59 (20.38)	<0.01
FVC (% predicted), Mean (SD)	86.33 (15.92)	90.92 (16.05)	0.01
Lowest FEV_1_ in the last 2 years (% predicted), Mean (SD)	65.51 (20.10)	65.33 (20.58)	0.91
**Laboratory findings**			
Current BEC (cells/μL), geom.mean (GSD) ^b^	203.34 ± 5.23	29.57 ± 13.93	<0.001
**Remission Criteria**			
ACT Score ≥ 20, *n* (%)	31 (33.7%)	190 (65.5%)	<0.001
No OCS, *n* (%)	79 (82.3)	246 (82.6)	1
FEV_1_ > 80% predicted, *n* (%)	26 (27.1)	132 (44.3)	<0.01
No Exacerbations, *n* (%)	35 (36.5)	178 (59.7)	<0.001
**Co-Morbidities**			
Allergies, *n* (%)	45 (48.4)	158 (53)	0.51
Sinusitis, *n* (%)	33 (35.5%)	143 (48.1%)	0.04
Nasal Polyps, *n* (%)	14 (15.1%)	117 (39.4%)	<0.001
GERD, *n* (%)	19 (20.4%)	79 (26.7%)	0.28
Depression, *n* (%)	13 (14%)	31 (10.4%)	0.45
COPD, *n* (%)	12 (12.9%)	30 (10.1%)	0.56
>2 LRTIs/year, *n* (%)	14 (15.1%)	44 (14.8%)	1

Abbreviations: ACT: Asthma Control Test; BEC: Blood Eosinophil Count; BMI: Body Mass Index; COPD: Chronic Obstructive Pulmonary Disease; GERD: Gastroesophageal Reflux Disease; ICS: Inhaled Corticosteroids; LRTI: Lower Respiratory Tract Infection; OCS: Oral Corticosteroids. BMI categories: Underweight BMI < 17.5 kg/m^2^; Normal Weight BMI 17.5–25 kg/m^2^; Overweight BMI 15–30 kg/m^2^; Obese > 30 kg/m^2^. ^b^ Geometric mean and geometric standard deviation (GSD) were used if the variable was not normally distributed.

## Data Availability

Data are available from the authors upon reasonable request.

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
