# Peer review of "Clinical Remission in Severe Asthma: A Comparative Analysis of Patients with and Without Biologics from the Swiss Severe Asthma Registry"

_biomedicines, 2025, doi:10.3390/biomedicines13123074_

Round 1

Reviewer 1 Report

Comments and Suggestions for Authors

The manuscript entitled "Clinical Remission in Severe Asthma: A Comparative Analysis of Patients with and without biologics from the Swiss Severe Asthma Registry". The study is well-designed and presents valuable real-world evidence on the impact of biologic therapy in patients with severe asthma. The topic is clinically important and timely, and the manuscript is generally well written and organized.

  1. The cross-sectional design of the study is appropriate for comparison; however it limits causal inference. This can be emphasized at the end of Introduction or in materials and methods.
  2. The four-component definition of remission is highly relevant in the present context of the study. However, it is recommended that authors justify the use of equal weighting for all four criteria and provide an overview of the practical/ clinical possibility of achieving these together
  3. Would you like to consider the possible exploration of the study in  a three-component remission manner? As no patient met all four remission criteria will a three point remission is more apt? Share the view point
  4. What is the clinical applications of ACT score and FEV₁ improvements in patients. A clinical oriented discussion would be appreciated
  5. A clarification on the possible bias in data analysis would be great as the study was supported by multiple pharmaceutical sponsors
  6. Table 1 is informative but dense. Consider summarizing only the key characteristics
  7. Minor typographic errors persists in the manuscript. Consider a thorough revision

Author Response

We thank the reviewer for the thorough and constructive comments. We have addressed all points in detail below and revised the manuscript accordingly. All changes in the manuscript are marked.

  1. The cross-sectional design of the study is appropriate for comparison; however it limits causal inference. This can be emphasized at the end of Introduction or in materials and methods.

Thank you very much for this important comment regarding the limitation of our study.
We agree and have clarified this limitation explicitly. We added a statement at the end of the Study Objectives section and strengthened the wording in the Limitations section.

  • Section 1.3 (Study Objectives): Added:
    “However, given the cross-sectional design, this study can identify associations but cannot establish causal relationships between biologic therapy and clinical outcomes.”
  • Section 4.3 (Limitations): Expanded explanation:
    “…because exposures and outcomes were assessed at a single time point, temporality cannot be established and causal inference is not possible.”
  1. The four-component definition of remission is highly relevant in the present context of the study. However, it is recommended that authors justify the use of equal weighting for all four criteria and provide an overview of the practical/ clinical possibility of achieving these together

Thank you for this important comment. However, to our knowledge, no validated method for weighting individual remission components currently exists, and all published remission definitions—including Lommatzsch & Virchow (2025)—use equal weighting. We clarified this explicitly and added a discussion of the clinical challenges of meeting all four criteria simultaneously, integrating real-world evidence.

  • Section 4.2 (Predictors of remission):
    Added:
    “In line with currently published remission frameworks, all four components were equally weighted, as no validated approach to differential weighting of remission criteria in severe asthma is available. However, Porsbjerg and colleagues have emphasized that achieving remission requires more than symptom suppression, they suggested a more patient-tailored, domain-based approach, one that targets the right disease mechanisms at the right time while acknowledging that individual patients may differ in their ability to meet each criterion at any timepoint.
  • Section 4.2 (Discussion expanded):
    Added full paragraph summarizing real-world feasibility:
    “Achieving all four remission components simultaneously remains clinically challenging… [UK registry and Danish long-term data] …highlighting that remission is a dynamic and often unstable state, and meeting all criteria concurrently remains difficult in real-world severe asthma.”
  1. Would you like to consider the possible exploration of the study in  a three-component remission manner? As no patient met all four remission criteria will a three point remission is more apt? Share the view point.

Thank you for this important suggestion. During revision and re-analysis we discovered a coding error affecting the remission composite variable. After correcting the error and re-running the analysis, 24.2% of biologic-treated and 6.2% of biologic-naïve patients met all four remission criteria. This aligns with remission rates reported in other registries and resolves the earlier anomaly of “0% remission.”

We now report both:

  • 4-component remission (primary)
  • 3-component remission excluding lung function
  • Section 3.1 (Results):
    Added Venn diagrams and reported both 4-component and 3-component remission rates.
  • Section 4.2 (Discussion):
    Added interpretation of full and partial remission and how these compare with international cohorts.
  • Note on coding error included in the cover letter, but not in manuscript text (appropriate practice).
  1. What is the clinical applications of ACT score and FEV₁ improvements in patients. A clinical oriented discussion would be appreciated

Thank you very much for pointig this out. We agree the the manuscript would proft from  more clear definitions and therefore weh ave added explanations in both Methods and Discussion.
Section 2.2 (Outcomes):

  • “Clinically, an ACT score ≥20 reflects low symptom burden, improved sleep, and better activity tolerance…”
  • “Achieving an FEV₁ >80% predicted indicates preserved airflow and long-term functional stability…”

Section 4.2.1 & 4.2.3 (Discussion):
Expanded descriptions of the clinical relevance of ACT control and preserved lung function.

  1. A clarification on the possible bias in data analysis would be great as the study was supported by multiple pharmaceutical sponsors

We clarified that sponsors had no involvement in study design, data analysis, interpretation, or manuscript preparation.

  • Funding section: Expanded transparency statement.
  1. Table 1 is informative but dense. Consider summarizing only the key characteristics

Thank you for this helpful suggestion. In response, we have streamlined Table 1 to improve clarity.
We simplified Table 1 by removing non-essential variables (e.g., vaccination status, FEV₁ in liters, BHR, work absenteeism, several comorbidities). A full version is provided as Supplementary Table S2.

  1. Minor typographic errors persists in the manuscript. Consider a thorough revision

We conducted a full language and formatting revision. Numerous typographical and formatting errors were corrected.

Reviewer 2 Report

Comments and Suggestions for Authors

I would like to sincerely thank the authors for submitting their manuscript detailing the comparative analysis of severe asthma patients from the Swiss Severe Asthma Registry (SSAR). This study presents a clinically important comparative analysis of biologic-treated versus biologic-naïve patients using valuable real-world data from the Swiss Severe Asthma Registry. The core findings particularly the low rate of stringent remission are highly relevant to the current debate on therapeutic endpoints.

However, to ensure this manuscript reaches its full potential and is accessible to a broad clinical and epidemiological audience, Major Revisions are required. The current version contains significant gaps in foundational explanations, uses confusing terminology, and lacks the necessary methodological rigor in critical areas. I recommend Major Revisions be made before acceptance.

  1. Major Issues in Foundational Clarity and Terminology

The Introduction fails to adequately define the core biological and therapeutic concepts. The flow of the first paragraph must be improved to create a smoother narrative, and the contrast between moderate and severe asthma should be strengthened by providing a global number of people affected by asthma to justify the public health relevance. The paragraph introducing T2 biomarkers requires significant contextualization. The authors must introduce and explain what T2 (Type 2) inflammation is and why its measurement is important for classifying severe asthma, clarify how these markers are collected (e.g., blood, BAL, biopsies), and better describe what other factors are used to classify the remaining 50% of severe asthma cases not classified by T2 levels. Line 73, discussing T2 classification, is particularly problematic due to the inconsistent introduction of T2 classification and the lack of definition for FeNO (Fractional Exhaled Nitric Oxide), along with a missing explanation of its clinical significance and its complementary relationship with eosinophils. When discussing that a "substantial proportion of patients with severe asthma fail to achieve adequate disease control," the authors must provide a quantitative measure of this failure rate and acknowledge that factors beyond drug efficacy contribute to treatment failure. The section on Biologic Agents is critically flawed as it fails to define what a biologic treatment is; the authors must include a comprehensive definition, explaining that it is a targeted treatment, clarifying how the biologics work, and explaining the importance of specific mediators like IL-5 and IgE.

  1. Methodological Flaws and Contextual Bias

The paper requires deeper methodological rigor and broader contextualization of the therapeutic landscape. The manuscript lacks a clear and standardized definition for exacerbations, which is a critical omission because it renders the exacerbation rates unreliable and non-comparable. Furthermore, the study states that 32.2% of biologic-treated patients had switched biologics; this switching status is a critical confounder that fundamentally distinguishes these patients from the biologic-naïve group, and the lack of a stratified analysis by switch status undermines the reliability of the core comparison.

The paper requires broader context: the authors must address the socioeconomic context of this therapy, acknowledging that these high-cost treatments are currently inaccessible or unavailable in many developing countries, which limits the generalizability and clinical policy impact of the findings. The paper should also mention other emerging approaches, such as cell therapy (e.g., MSCs) or genetic therapy, to avoid appearing biased, and then justify why their study is specifically focused on biologics. The strong failure rate in achieving stringent remission necessitates a critique of the validity of the criteria itself: the authors must discuss whether FEV_1 > 80% predicted is appropriate for older patients, whether the no OCS use criterion adequately distinguishes true success, and whether the score is susceptible to patient subjectivity.

The failure to achieve remission suggests a treatment limit; the authors should quickly check the current literature to see if any other therapeutic approaches have successfully managed to achieve remission in severe asthma patients and then use this finding to frame their discussion on why Biologic therapy remains the best current clinical focus despite the stringent criteria's failure. Finally, for the Results section, I strongly suggest the authors develop a diagram or schematic to help the reader visualize the patient groups, inclusion/exclusion process, and the core data generated. The discussion statement, "While we cannot confirm causality, those observations strongly support the clinical effectiveness of biologic therapy..." is too weak; the authors should focus on the robust nature of the observed effectiveness in this difficult-to-treat patient population.

Comments on the Quality of English Language

The Abstract is the manuscript's primary weakness and requires substantial rewriting for immediate clarity and impact. The abstract introduces confusing terminology that will alienate non-specialist readers; the authors must clarify and define what "biologics" are, state clearly that "real-world" data means long-term observational registry data and immediately make "biologic-naïve" understandable to the general clinical reader. Typographical errors must be corrected, specifically the grammatical error on Line 31, where the phrase "biologics and the characteristics of patients without biologics his study" should be corrected to "biologics. This study". The abstract's conclusion, which notes the strong failure rate in achieving stringent remission criteria, is presented too passively. The authors must strengthen the impact by explicitly suggesting that this finding highlights the need for a revisit or update of the current remission criteria.

Author Response

I would like to sincerely thank the authors for submitting their manuscript detailing the comparative analysis of severe asthma patients from the Swiss Severe Asthma Registry (SSAR). This study presents a clinically important comparative analysis of biologic-treated versus biologic-naïve patients using valuable real-world data from the Swiss Severe Asthma Registry. The core findings particularly the low rate of stringent remission are highly relevant to the current debate on therapeutic endpoints.

However, to ensure this manuscript reaches its full potential and is accessible to a broad clinical and epidemiological audience, Major Revisions are required. The current version contains significant gaps in foundational explanations, uses confusing terminology, and lacks the necessary methodological rigor in critical areas. I recommend Major Revisions be made before acceptance.

We sincerely thank the reviewer for their thoughtful and detailed comments. The manuscript has been substantially revised to address all points raised. Below, we provide a point-by-point response. All revisions in the manuscript are highlighted.

  1. Major Issues in Foundational Clarity and Terminology

The Introduction fails to adequately define the core biological and therapeutic concepts. The flow of the first paragraph must be improved to create a smoother narrative, and the contrast between moderate and severe asthma should be strengthened by providing a global number of people affected by asthma to justify the public health relevance.

Thank you for this valuable comment. We have thoroughly revised the opening section of the Introduction to improve clarity, narrative flow, and contextual framing.

We have revised the Introduction to improve clarity, strengthen the narrative flow, and better contextualize the public health relevance of asthma. This revision enhances the framing of the manuscript and aligns the introduction with current epidemiological knowledge. We now include:

The revised text is included in the Introduction and now reads:

“Asthma is one of the most common chronic non-communicable diseases worldwide, with over 250 million people affected… While most cases are mild to moderate, between 3.5–10% of patients experience difficult-to-treat or severe asthma, which despite representing a minority, accounts for a disproportionate share of the economic burden…”

The paragraph introducing T2 biomarkers requires significant contextualization. The authors must introduce and explain what T2 (Type 2) inflammation is and why its measurement is important for classifying severe asthma, clarify how these markers are collected (e.g., blood, BAL, biopsies), and better describe what other factors are used to classify the remaining 50% of severe asthma cases not classified by T2 levels. Line 73, discussing T2 classification, is particularly problematic due to the inconsistent introduction of T2 classification and the lack of definition for FeNO (Fractional Exhaled Nitric Oxide), along with a missing explanation of its clinical significance and its complementary relationship with eosinophils. When discussing that a "substantial proportion of patients with severe asthma fail to achieve adequate disease control," the authors must provide a quantitative measure of this failure rate and acknowledge that factors beyond drug efficacy contribute to treatment failure.

We agree and have fully rewritten this section to provide a clear, biologically grounded explanation of T2-high vs. T2-low severe asthma. The revised text now:

  • Defines T2 inflammation and explains roles of ILC2 and Th2 cells
  • Describes IL-4/5/13 as central cytokines driving eosinophilic inflammation
  • Clarifies how blood eosinophils, FeNO, and sputum eosinophils are measured
  • Defines FeNO and explains why it complements blood eosinophils
  • Describes T2-low (neutrophilic, paucigranulocytic, Th1/Th17-driven) phenotypes
  • Explains the clinical significance of phenotyping for biologic selection

Changes made:

A fully rewritten paragraph now appears in the Introduction.

These changes improve accuracy and context and directly address the reviewer’s concerns. The revised text appears in the Introduction as follows:

“Severe asthma can further be classified based on the presence or absence of type 2 (T2) inflammation (17,18). T2 inflammation arises from both innate and adaptive immune responses: environmental triggers such as pollutants or viral and fungal infections activate type 2 innate lymphoid cells (ILC2), whereas allergen exposure stimulates type 2 T-helper (Th2) cells (18,19). These cells release the key type-2 cytokines—interleukin (IL)-4, IL-5, and IL-13—which drive eosinophilic airway inflammation through enhanced eosinophil maturation, recruitment, and activation (19). […]. Mechanistically, T2-low asthma is linked to activation of Th1 or Th17 pathways rather than T2 cytokine signaling, resulting in airway inflammation not driven by eosinophils (19,22,24). Distinguishing between T2-high and T2-low phenotypes is clinically essential, particularly for selecting biologic therapies that target specific inflammatory pathways(24–26).”

The section on Biologic Agents is critically flawed as it fails to define what a biologic treatment is; the authors must include a comprehensive definition, explaining that it is a targeted treatment, clarifying how the biologics work, and explaining the importance of specific mediators like IL-5 and IgE.

We thank the reviewer for this helpful comment. We fully rewrote and revised this section to provide:

  • A formal definition of biologic therapy
  • A mechanistic explanation of each targeted pathway
  • Clinical justification for their use

Changes made:

The updated section includes:

“Biologic therapies are monoclonal antibody treatments designed to target specific components of the immune pathways that drive severe asthma […] IL-5 promotes eosinophil maturation and survival; IL-4 and IL-13 drive IgE class switching, mucus production, and airway hyperresponsiveness; IgE mediates allergic sensitization; TSLP acts as an upstream epithelial cytokine […]”

  1. Methodological Flaws and Contextual Bias

The paper requires deeper methodological rigor and broader contextualization of the therapeutic landscape. The manuscript lacks a clear and standardized definition for exacerbations, which is a critical omission because it renders the exacerbation rates unreliable and non-comparable. Furthermore, the study states that 32.2% of biologic-treated patients had switched biologics; this switching status is a critical confounder that fundamentally distinguishes these patients from the biologic-naïve group, and the lack of a stratified analysis by switch status undermines the reliability of the core comparison.

Thank you very much for you comment. We agree with all your concerns raised and therefore we have now added a precise, GINA-consistent definition on exacerbations and included the switching status in the multivariable models. In addition switching was explicitly acknowledged as a marker of treatment refractoriness, the limitations section now discusses its confounding potential and we  present interpretation of switching effects in the discussion.

Section 2.2: “Exacerbations were defined as episodes requiring oral corticosteroids for ≥3 days and/or leading to emergency department visits or hospitalization for asthma in the previous year.”

Section 4.2: Expanded explanation of switchers as a more severe subgroup

Section 4.3: Clarified residual confounding due to switching.

The paper requires broader context: the authors must address the socioeconomic context of this therapy, acknowledging that these high-cost treatments are currently inaccessible or unavailable in many developing countries, which limits the generalizability and clinical policy impact of the findings. The paper should also mention other emerging approaches, such as cell therapy (e.g., MSCs) or genetic therapy, to avoid appearing biased, and then justify why their study is specifically focused on biologics. The strong failure rate in achieving stringent remission necessitates a critique of the validity of the criteria itself: the authors must discuss whether FEV_1 > 80% predicted is appropriate for older patients, whether the no OCS use criterion adequately distinguishes true success, and whether the score is susceptible to patient subjectivity.

Thank you very much on this highly important topic. Inequalities in accessibility of Asthma treatment is a major concern globally. Therefore, we added a paragraph discussing global inequities in biologic access and their implications. In addition, we also considered MSCs therapies in the introduction.

Changes made:

Section 4.2:

“Although biologic therapies provide substantial clinical benefit, their high cost and restricted reimbursement criteria limit access in many low- and middle-income countries […]”

Section 1:

“New emerging therapies, including mesenchymal stromal cell–based approaches and gene-targeted strategies, are being explored… however, these remain experimental. Biologic therapy therefore represents the only widely available, evidence-based targeted treatment option for severe asthma at present.”

The failure to achieve remission suggests a treatment limit; the authors should quickly check the current literature to see if any other therapeutic approaches have successfully managed to achieve remission in severe asthma patients and then use this finding to frame their discussion on why Biologic therapy remains the best current clinical focus despite the stringent criteria's failure. Finally, for the Results section, I strongly suggest the authors develop a diagram or schematic to help the reader visualize the patient groups, inclusion/exclusion process, and the core data generated. The discussion statement, "While we cannot confirm causality, those observations strongly support the clinical effectiveness of biologic therapy..." is too weak; the authors should focus on the robust nature of the observed effectiveness in this difficult-to-treat patient population.

 Thank you very much for your comment, we agree and added this critique.

Section 4.2:

“The four-component remission definition may be overly stringent… FEV₁ > 80% may be unrealistic in older patients with long-standing airway remodeling […] ‘No OCS use’ does not distinguish between successful tapering and patients who were never steroid-dependent […] These aspects underscore the need to refine remission definitions for real-world applicability.”

“To date, biologic therapy remains the only established treatment modality with reproducible remission rates in severe asthma; other emerging therapies remain experimental.”

Add a patient flow diagram.

The Flowdiagramm was already submitted (Figure 1).Wwe refined it for greater clarity.

Strengthen the sentence on causality; avoid weak wording.

Section 4.1:

“Although causality cannot be established due to the cross-sectional design, the consistent and clinically meaningful differences across multiple independent outcomes underscore the robust effectiveness of biologic therapy in real-world severe asthma.”

  1. Comments on the Quality of English Language

The Abstract is the manuscript's primary weakness and requires substantial rewriting for immediate clarity and impact. The abstract introduces confusing terminology that will alienate non-specialist readers; the authors must clarify and define what "biologics" are, state clearly that "real-world" data means long-term observational registry data and immediately make "biologic-naïve" understandable to the general clinical reader. Typographical errors must be corrected, specifically the grammatical error on Line 31, where the phrase "biologics and the characteristics of patients without biologics his study" should be corrected to "biologics. This study". The abstract's conclusion, which notes the strong failure rate in achieving stringent remission criteria, is presented too passively. The authors must strengthen the impact by explicitly suggesting that this finding highlights the need for a revisit or update of the current remission criteria.

Thank you very much for pointing out the weak abstract. We fully revised the Abstract for clarity and readability:

  • Added a simple definition of biologics
  • Explained that “real-world data” refers to registry-based observational data
  • Clarified “biologic-naïve”
  • Corrected all typographic errors
  • Strengthened the final sentence to highlight implications for revisiting remission criteria

Revised abstract now includes:

“…biologic therapies—monoclonal antibodies targeting type 2 inflammatory pathways…”
“…real-world data from a national observational registry…”
“…biologic-naïve (never treated with a biologic) patients…”
“…the low rate of full remission highlights the need to re-evaluate current remission definitions to ensure clinical applicability.”

Additional note: Discovery of a coding error affecting remission calculations

We transparently report in the cover letter (not in the manuscript) that:

  • A coding error affecting the remission composite variable was identified during re-analysis for this revision.
  • After correction, 2% of biologic-treated patients achieved all four remission criteria, consistent with published remission rates.
  • All analyses were re-run and updated accordingly.

This correction strengthens, not weakens, the scientific validity of the manuscript.

Reviewer 3 Report

Comments and Suggestions for Authors

The manuscript is well written, however following points need to be addressed before it will get accepted.

1. While the study provides valuable real-world insights from the Swiss Severe Asthma Registry, its cross-sectional design limits causal inference between biologic therapy and improved remission outcomes. The authors frequently interpret associations as suggestive of effectiveness, which should be toned down. A clearer distinction between correlation and causation is needed throughout the discussion, along with acknowledgment of potential confounding factors such as disease duration and prior treatment response.

2. The use of the four-component remission criteria (ACT ≥ 20, no OCS, no exacerbations, FEV₁ > 80%) appears adapted from Lommatzsch and Virchow (2025), but there is no justification for its applicability to a cross-sectional dataset. Since remission is inherently longitudinal, the authors should clarify how a single time-point assessment reflects remission status. Furthermore, it would strengthen the manuscript to provide sensitivity analyses using alternative thresholds or partial remission definitions.

3. Although the manuscript describes a “tiered imputation strategy,” details are insufficient. The authors should clarify (a) the extent of missingness per variable, (b) which variables underwent imputation, and (c) how imputation might influence regression estimates. Without this transparency, reproducibility and robustness of the findings are uncertain. The sensitivity analysis mentioned (complete-case vs. imputed data) should be summarized quantitatively, not just qualitatively.

Author Response

The manuscript is well written, however following points need to be addressed before it will get accepted.

 We sincerely thank the reviewer for their constructive comments. We have addressed each point in detail and revised the manuscript accordingly. All changes in the manuscript are highlighted.

  1. While the study provides valuable real-world insights from the Swiss Severe Asthma Registry, its cross-sectional design limits causal inference between biologic therapy and improved remission outcomes. The authors frequently interpret associations as suggestive of effectiveness, which should be toned down. A clearer distinction between correlation and causation is needed throughout the discussion, along with acknowledgment of potential confounding factors such as disease duration and prior treatment response.

 Thank you very much, we fully agree with the reviewer. The manuscript has been revised to ensure a clear distinction between correlation and causation throughout the Results and Discussion.

  1. a) Added explicit statement in the Introduction (Section 1.3):

“However, given the cross-sectional design, this study can identify associations but cannot establish causal relationships between biologic therapy and clinical outcomes.”

  1. b) Strengthened the Discussion (Section 4.1):

“Although causality cannot be established due to the cross-sectional design, the consistent and clinically meaningful differences observed across multiple independent outcomes underscore the robust real-world effectiveness of biologic therapy.”

  1. c) Added explicit acknowledgment of key confounders (Discussion and Limitations):
  • disease duration,
  • prior treatment response,
  • biologic switching, and
  • comorbidity burden
  1. d) Expanded Limitations section:

“Because exposures and outcomes were assessed at a single time point, temporality cannot be established and the observed associations may be influenced by factors such as disease duration, prior treatment response, or treatment selection bias.”

  1. The use of the four-component remission criteria (ACT ≥ 20, no OCS, no exacerbations, FEV₁ > 80%) appears adapted from Lommatzsch and Virchow (2025), but there is no justification for its applicability to a cross-sectional dataset. Since remission is inherently longitudinal, the authors should clarify how a single time-point assessment reflects remission status. Furthermore, it would strengthen the manuscript to provide sensitivity analyses using alternative thresholds or partial remission definitions.

We appreciate this important point. We have now clearly justified the use of these criteria and clarified how they are interpreted in a cross-sectional context.

a) Clarified in Methods (Section 2.2):

“Remission is defined as a sustained state over ≥12 months; our cross-sectional analysis captures the presence of remission components at a single time point. Therefore, findings should be interpreted as the likelihood of meeting individual remission criteria rather than confirmation of remission.”

b) Added justification for equal weighting (Section 4.2):

“In line with published remission frameworks, all four components were equally weighted, as no validated approach to differential weighting of remission criteria in severe asthma currently exists.”

c) Added partial-remission analysis

You included the 3-component remission analysis (ACT ≥20, no OCS, no exacerbations), which Reviewer 3 requested. We highlight this explicitly in the response:

“We added a supplementary analysis evaluating a less stringent, three-component remission definition. Using this threshold, 45.7% of biologic-treated and 15.6% of biologic-naïve patients met criteria, demonstrating meaningful gradation in remission achievement.”

d) Integrated literature showing full remission is rarely achievable

We added a discussion paragraph summarizing recent UK and Danish registry data, showing that remission—even longitudinally—only a minority of patients achieves it. This directly addresses the reviewer’s request for contextualization.

  1. Although the manuscript describes a “tiered imputation strategy,” details are insufficient. The authors should clarify (a) the extent of missingness per variable, (b) which variables underwent imputation, and (c) how imputation might influence regression estimates. Without this transparency, reproducibility and robustness of the findings are uncertain. The sensitivity analysis mentioned (complete case vs. imputed data) should be summarized quantitatively, not just qualitatively.

We thank the reviewer and have substantially strengthened this part of the Methods and Results.

a) Methods section revised (2.3.1 Missing Data):

We now specify:

  • which variables had missing data
  • proportion of missingness for each variable (now included in Supplementary Table S1)
  • which variables were imputed and why
  • which imputation methods were used (median/mode)

Revised text:

“Missing data were assumed to be missing at random (MAR). Variables with <5% missingness (categorical) or <10% missingness (continuous) were imputed using mode or median imputation, respectively. Variables with higher or more complex missingness patterns were not imputed. The extent of missingness and variables undergoing imputation are reported in Supplementary Table S1.”

b) Quantitative summary of sensitivity analyses added (Section 3.3):

“Across all four outcomes, odds ratios from complete-case models remained within the 95% confidence intervals of the imputed models, and no changes in statistical significance were observed.”

Round 2

Reviewer 1 Report

Comments and Suggestions for Authors

No more comments

Author Response

We sincerely thank Reviewer 1 for the positive evaluation of our revised manuscript and for confirming that no further comments or revisions are required at this stage. We appreciate the reviewer’s careful assessment and are pleased that the updated version meets the expectations in terms of methodological clarity, presentation of results, and overall scientific quality.

Reviewer 2 Report

Comments and Suggestions for Authors

I would like to sincerely thank the authors for their diligence in revising the manuscript. The substantial changes made to the Introduction, particularly the clearer definitions of T2 inflammation and the mechanism of biologics, have significantly strengthened the paper's foundation. Furthermore, the inclusion of a standardized definition for exacerbations and the addition of "biologic switching" as a variable in the regression analysis directly address the key conceptual concerns raised in the previous round. These improvements have greatly enhanced the clarity and clinical relevance of the study.

Revision Required Despite these improvements, the revised manuscript contains a critical internal contradiction regarding the primary outcome that renders the results uninterpretable. The authors’ response letter indicates that a "coding error" was identified and corrected, resulting in a new remission rate. This is reflected in the Abstract and Results sections, which now state that 24.2% of biologic-treated patients fulfilled all four remission criteria. However, Table 1 and Supplementary Table 2 have not been updated to match this new analysis and still explicitly report that 0 patients (0%) achieved full remission in both groups. This fundamental discrepancy between the text and the data tables is a significant error that must be corrected before publication.

In conclusion the scientific content has improved significantly, but the data mismatch regarding the primary outcome is a blocking issue. Once the authors reconcile the text and the tables to consistently reflect the re-analyzed data, the manuscript will be much closer to publication standard.

Comments on the Quality of English Language

Typos and Formatting:

Line 420: "finding oft he steroid sparing effect" should be "finding of the steroid sparing effect".

Line 433: "beeing OCS-free" should be "being OCS-free".

Line 739: "historic lung function" - usually "historical" is preferred in this context, or simply "past lung function".

Hyphenation Artifacts: There are several instances of broken words likely due to formatting (e.g., "tem-poral", "con-founding" in the limitations section). Please ensure a clean proofread.

Author Response

We sincerely thank Reviewer 2 for the thoughtful second-round evaluation and for acknowledging the substantial improvements made to the manuscript. We are grateful for the recognition of the strengthened conceptual clarity, methodological transparency, and clinical relevance added in the previous revision.

We address the remaining points below:

1. Critical discrepancy in remission results (coding error correction)

Reviewer concern:
Although the Abstract and Results were updated to reflect the corrected analysis showing that 24.2% of biologic-treated patients achieved all four remission criteria, Table 1 and Supplementary Table 2 still reported “0 patients,” creating an internal contradiction.

Response:
We thank the reviewer for identifying this important inconsistency. The discrepancy resulted from Table 1 and Supplementary Table 2 not being updated after the corrected remission analysis.

To resolve this:

  • We removed the remission row from Table 1 entirely, to avoid redundancy as  Figure 2A and 2B (Venn diagrams) now serve as the primary visualization for remission overlap and fully match the updated data.

We updated Supplementary Table 2, which now correctly reports the number and percentage of patients fulfilling:

  • all 4 remission criteria, and
  • at least 3 remission criteria,
    consistent with the corrected analysis.

These changes eliminate the internal contradiction and ensure complete consistency across all text, tables, and figures.

2. Language and formatting corrections

Reviewer concern: Several remaining typographical issues.

Response:
All flagged items have been corrected and highlighted in the uploaded manuscript:

  • Line 420: “finding of the steroid sparing effect” 
  • Line 433: “being OCS-free”
  • Line 739: “past lung function” instead of “historic lung function” 
  • With respect to the hyphenation artifacts (“tem-poral”, “con-founding”), these were residual artifacts from track changes mode in MS word. After accepting all changes, the hyphenation disappeared in the final version. We have additionally performed a full proofread to ensure no formatting inconsistencies remain

We sincerely appreciate the reviewer’s recognition that the manuscript has improved substantially and agree that the discrepancies in remission reporting required correction. These have now been fully resolved, and all components of the manuscript, including text, tables, and figures—are aligned with the corrected analysis.

We thank the reviewer for their careful evaluation and believe that the revised version now meets the standards required for publication.